# Experimental In-Field Transfer and Survival of *Escherichia coli* from Animal Feces to Romaine Lettuce in Salinas Valley, California

**DOI:** 10.3390/microorganisms7100408

**Published:** 2019-09-29

**Authors:** Saharuetai Jeamsripong, Jennifer A. Chase, Michele T. Jay-Russell, Robert L. Buchanan, Edward R. Atwill

**Affiliations:** 1Western Center for Food Safety, University of California, Davis, CA 95618, USAmjay@ucdavis.edu (M.T.J.-R.); 2Research Unit in Microbial Food Safety and Antimicrobial Resistance, Department of Veterinary Public Health, Faculty of Veterinary Science, Chulalongkorn University, Bangkok 10330, Thailand; 3Center of Food Safety and Security Systems, College of Agricultural and Natural Resources, University of Maryland, MD 20742, USA; rbuchana@umn.edu

**Keywords:** agriculture, *E. coli* (all potentially pathogenic types), EHEC (enterohaemorrhagic *E. coli*), food safety, irrigation, microbial contamination, produce, Romaine lettuce

## Abstract

This randomized controlled trial characterized the transfer of *E. coli* from animal feces and/or furrow water onto adjacent heads of lettuce during foliar irrigation, and the subsequent survival of bacteria on the adaxial surface of lettuce leaves. Two experiments were conducted in Salinas Valley, California: (1) to quantify the transfer of indicator *E. coli* from chicken and rabbit fecal deposits placed in furrows to surrounding lettuce heads on raised beds, and (2) to quantify the survival of inoculated *E. coli* on Romaine lettuce over 10 days. *E. coli* was recovered from 97% (174/180) of lettuce heads to a maximal distance of 162.56 cm (5.33 ft) from feces. Distance from sprinklers to feces, cumulative foliar irrigation, and lettuce being located downwind of the fecal deposit were positively associated, while distance from fecal deposit to lettuce was negatively associated with *E. coli* transference. *E. coli* exhibited decimal reduction times of 2.2 and 2.5 days when applied on the adaxial surface of leaves within a chicken or rabbit fecal slurry, respectively. Foliar irrigation can transfer *E. coli* from feces located in a furrow onto adjacent heads of lettuce, likely due to the kinetic energy of irrigation droplets impacting the fecal surface and/or impacting furrow water contaminated with feces, with the magnitude of *E. coli* enumerated per head of lettuce influenced by the distance between lettuce and the fecal deposit, cumulative application of foliar irrigation, wind aspect of lettuce relative to feces, and time since final irrigation. Extending the time period between foliar irrigation and harvest, along with a 152.4 cm (5 ft) no-harvest buffer zone when animal fecal material is present, may substantially reduce the level of bacterial contamination on harvested lettuce.

## 1. Introduction

Fresh fruits and vegetables are now recognized worldwide as a source of the foodborne transmission of zoonotic pathogens [1,2,3]. Shiga toxin-producing *E. coli* O157:H7 is an enteric pathogen that poses severe risk to public health when present in food due to its low infectious dose and high pathogenic potential to cause hemorrhagic colitis and hemolytic uremic syndrome [4]. In 1995, the first reported outbreak of pathogenic *E. coli* O157:H7 linked to lettuce contamination was documented in Montana, although the source of the contamination was not identified [5]. In 1996, a multistate outbreak of *E. coli* O157:H7 infection associated with mesclun lettuce was identified in Connecticut and Illinois where wash water was the suspected source of the contamination [6]. Since these early reports of lettuce-related *E. coli* O157:H7 outbreaks, additional commodities of fresh fruits and vegetables have been identified as sources of foodborne pathogens such as strawberries, unpasteurized apple cider, lettuce, spinach, alfalfa sprouts, and bagged mixed salad [6,7,8,9,10,11,12]. Consumption of spinach or lettuce, for example, was associated with over 20 outbreaks and accounted for nearly half of all produce outbreaks during the period between 1995 to 2006 [13,14].

Domesticated animals and wildlife are important reservoirs of *E. coli* O157:H7 [15,16]. However, the exact mechanism and indigenous sources (root cause) of leafy greens and other produce-related *E. coli* O157:H7 outbreaks have rarely been identified. At the pre-harvest stage, potential sources of contamination include irrigation and other sources of agriculture water, soil amended with untreated manure, fecal deposits from domestic and wild animals intruding into produce fields, and infected farm workers handling and/or processing produce [17,18,19]. Proximity and contact of irrigation and surface waters with livestock and wildlife feces, animal manure matrices, and improperly used or untreated manure have been identified as risk factors for pre-harvest contamination of leafy greens that could lead to foodborne transmission to consumers [20,21]. 

Wildlife intrusion followed by defecation in produce production fields can readily introduce zoonotic pathogens into the produce growing and harvesting environment [22,23]. Animal scat can contaminate produce through fecal splash due to kinetic energy from foliar irrigation systems [20]. Previous research into *E. coli* O157:H7 contamination of lettuce and its survival has primarily focused on contaminated water and soil-amended manure [18,24,25,26,27]. Recent leafy green field trials in the western and the northeastern United States have studied the transfer and survival of foodborne pathogen surrogates from animal fecal material onto lettuce plants [20,28,29,30,31].

The United States Food and Drug Administration (U.S. FDA) Food Safety Modernization Act (FSMA) Produce Safety Rule addresses risks from wildlife entering fields or packing areas in Subpart I of the regulation [32]. The Rule requires that “if there is reasonable probability that animal intrusion will contaminate produce, that those areas used for growing covered produce are monitored as needed during the growing season and immediately prior to harvest.” Specifically, produce covered under the rule cannot be harvested if visibly contaminated or likely to be contaminated with animal feces.

The California and Arizona Leafy Greens Marketing Agreement (LGMA), an industry-driven food safety program, contains mandatory guidelines for produce food safety practices for lettuce, spinach, and other leafy green commodities by focusing in part on soil management, proximity to livestock operations, agricultural water quality, wildlife intrusion and other environmental conditions together with farm worker hygiene. The LGMA requires a minimum five-foot radius no-harvest buffer zone surrounding a fecal deposit to reduce the risk of contamination from fecal-borne zoonotic pathogens [33]. However, there is limited information regarding the validity of this no-harvest buffer zone metric.

The purpose of this randomized controlled field trial was several-fold. Our first goal was to measure the transfer of *E. coli* from fresh fecal material onto nearby heads of lettuce when foliar or overhead sprinkler irrigation was applied. Second, we wanted to identify key management and other environmental factors that could impact the magnitude of bacterial transfer onto lettuce surfaces in an agricultural field including the use of a five-foot radius buffer between a fecal deposit and harvested heads of lettuce. Third, we wanted to estimate the rate of environmental inactivation of indicator *E. coli* as a function of animal fecal slurry type when applied onto the adaxial surface of lettuce, and how foliar irrigation might influence the inactivation or growth kinetics. Under conventional growing practices (bed-furrow configuration, irrigation design, etc.), this study developed a statistical model of risk factors for pathogen transference onto heads of lettuce that estimates the mean bacterial contamination levels following foliar irrigation on lettuce heads.

## 2. Materials and Methods 

### 2.1. Lettuce Field Conditions

Romaine lettuce (*Lactuca sativa* L. var. *Longifolia*) was grown, thinned, and weeded following standard agricultural practices by the United States Department of Agriculture’s Agricultural Research Service (USDA-ARS) in the South Salinas Valley, Monterey County, California from June through August 2012. One Romaine lettuce field contained ten wide beds with approximately 0.4 m wide furrows as described previously [21]. Two rows of seeds were planted approximately 0.3 m apart in 0.6 m wide beds. Overhead sprinklers using 46 cm (18 in) risers and Nelson rotary nozzles were spaced around the field in a square pattern (9.12 by 9.12 m or 30 by 30 ft). The leafy growing field was 44 m long, with 12 overhead sprinklers. The lettuce field was irrigated for about two hours with groundwater every five to seven days. These growing and irrigation practices, along with the field design, are typical of commercial operations in the region. For Experiment A, irrigation was suspended for five days before the start of the trial, with the next irrigation on day 0 (6 August 2012) commencing at 8.30 a.m. for around 2.5 h. In Experiment B, fecal slurries were applied onto the adaxial surface of Romaine lettuce leaves immediately after the irrigation event on 1 August 2012. All pesticide application was discontinued at least three days before inoculation.

Environmental factors such as cumulative precipitation (cm), ambient air temperature (°C), relative humidity (%), wind direction (0–360), and mean wind speed 24 h prior to harvest (mph) were retrieved from the closest station of the California Irrigation Management Information System (CIMIS) from June to August, 2012 (www.tid.org/water/water-management/california-irrigation-management-information-system-cimis). In-field potential contributing factors included the average distance from the four closest sprinklers to the fecal deposit (cm); the distance from each lettuce head to fecal deposit (cm); lettuce orientation relative to wind direction and fecal deposit (upwind or downwind of fecal deposit); age of inoculated feces prior to irrigation (days); and vertebrate source of feces (chicken or rabbit). These distances defined above were measured approximately two weeks before starting the field trial. After harvesting all lettuce for *E. coli* determination, the application of foliar irrigation (cm) was measured using two rain gauges placed within each cluster of eight lettuce heads.

### 2.2. Bacterial Strain and Inoculum Preparation

Indicator *E. coli* strain (TVS 354) resistant to rifampicin at a concentration of 100 µg/mL (*E. coli* Rif^r^) was isolated originally from Romaine lettuce in the Salinas Valley (Trevor Suslow, Division of Agricultural and Natural Resources, UC Davis, Davis, USA) and used as a surrogate of *E. coli* O157:H7. The *E. coli* Rif^r^ stock culture was streaked for individual colonies onto tryptic soy agar (TSA; Difco, Becton Dickinson, Sparks, MD, USA) plates supplemented with 100 µg/mL of rifampicin (Rif; Golden Biotechnology, St. Louis, MO, USA) (TSA-Rif) and incubated at 37 °C overnight. The bacterial strain was stored in a sterile cryo-bead tube and kept at −80 °C until required. One single bead was utilized for inoculation into trypticase soy broth (TSB; Difco, BD, USA) supplemented with 100 µg/mL of rifampicin (TSB-Rif), and grown with agitation at 37 °C for 5 to 6 h under orbital rotation (100 rpm) until the cell density was approximately 10^8^ to 10^9^ cells/mL to achieve a final inoculum cell density of 10^7^ to 10^8^ cells/g feces. The stock suspension concentration was estimated using a spectrophotometer (BioMate 3, Thermo Spectronic, Rochester, NY, USA) at 600 nm according to a standard growth curve prior to use. The concentration was confirmed by plating serial dilutions in phosphate buffered saline (PBS; Sigma-Aldrich, St. Louis, MO, USA) onto triplicate TSA-Rif agar plates.

### 2.3. E. coli Transfer (Experiment A)

Using procedures described by Atwill et al. [21], 50 g of fresh laboratory rabbit and chicken feces were collected from sources provided by the Department of Animal Science, UC Davis and tested the presence of *E. coli* Rif^r^ and *E. coli* O157:H7. All fecal samples were found to be negative for both *E. coli* Rif^r^ and *E. coli* O157:H7, indicating that our source of fecal material was either truly negative or below our detection limit of less than 0.1 CFU/g. Fresh feces were stored at 4 °C and inoculated with prepared *E. coli* Rif^r^ stock solution within 24 h after fecal collection. Rabbit and chicken feces were separately combined with 50 and 5 mL of PBS spiked with the target inoculum of *E. coli* Rif^r^, respectively. Each species of animal feces and *E. coli* Rif^r^ inoculum were mixed using a stomacher (Stomacher 400 circulator paddle blender, Seward Worthington, UK) at 250 rpm for 5 min, with the goal of achieving a final bacterial concentration of 10^7^ to 10^8^ cells/g feces. After homogenization, four 5 g subsamples of each inoculated fecal aliquot (fecal deposit) were wrapped in Saran wrap and kept at 4 °C until deposited in the field, within 12 h of preparation.

The locations of twenty-three clusters of Romaine lettuce were randomly selected across 10 adjacent beds of commercially grown lettuce, where one cluster was composed of eight heads of mature Romaine lettuce equally divided between two adjacent beds on each side of the furrow and a fecal deposit placed in the middle of the furrow (Figure 1). Within each cluster, we stratified the distance between the eight heads of lettuce and the fecal deposit to achieve a balanced sample size of different distances between the lettuce and feces (Figure 1). The maximum distance between the feces and heads of lettuce was dictated by the LGMA recommendation of a minimum radius of 152.4 cm (5 ft) for the no-harvest buffer zone when fecal deposits are observed in a lettuce field intended for human consumption. The resulting 184 heads of lettuce, along with eight lettuce negative controls, were identified by using labeled polyvinyl chloride flags placed behind the lettuce heads two weeks before the start of the experiment. The order of rabbit and chicken fecal deposits was also randomized and simulated wild rabbit and wild bird defecation in the furrows of a field of mature Romaine lettuce near the time of harvest and between four days to several hours (days −4, −2, −1, 0) prior to irrigation on day 0 (Figure 1). The field inoculation started from 2 August (day −4) to 6 August (day 0) 2012 between 8 and 9 a.m. with the bacterial inoculum, and chicken and rabbit fecal deposits prepared within 12 h of each inoculation day at UC Davis. Two rain gauges were placed in the furrow flanking the fecal deposit, approximately one foot away, to measure the total application of irrigation water. Immediately before placement of the fecal deposits for each inoculation day, the concentration of *E. coli* Rif^r^ in three 1 g aliquots of inoculated chicken and rabbit feces was determined by serial dilution in PBS and plating in triplicate on TSA-Rif plates at the field site using a mobile laboratory.

### 2.4. Fecal Slurry Inoculation onto the Upper Surface of Lettuce Leaves (Experiment B)

Following the procedures described by Chase et al. [28,34], 125 g of chicken and rabbit feces were collected and separately mixed with 500 mL PBS in a stomacher bag and stomached for 5 min at 250 rpm. Fecal slurries, void of fecal fiber or debris, were transferred to sterile plastic cups then mixed with of *E. coli* Rif^r^ for a final concentration between 10^7^ to 10^8^ CFU/mL. The slurry suspensions were kept at 4 °C and transported to the field site in a portable cooler. The slurry concentrations were confirmed using 10-fold serial dilutions and plating in triplicate onto TSA-Rif agar plates, immediately after preparation, and then again immediately prior to field inoculations. 

One hundred and ninety-two heads of lettuce were randomly selected across 20 beds two weeks prior to the start of the field trial with colored flags placed near each selected head to signify either of two species of feces to be applied. In addition, 44 negative controls were randomly selected at the end of the field to minimize the chance for cross-contamination from inoculated lettuce. Rabbit and chicken fecal slurries were mixed to re-homogenize the solution and then 500 µL was inoculated directly onto the upper (adaxial) surface of the second-most outer leaf of Romaine lettuce on 1 August 2012 (day 0) from 8 to 10 a.m. Enumeration of surviving bacteria/head of lettuce was conducted each day for 10 additional days until 11 August 2012 (day 10). Eight heads of lettuce were collected per day for each animal species of slurry (*n* = 16), except for day 5 when the lettuce head samples were harvested immediately before and after sprinkler irrigation (*n* = 32). Lettuce samples were harvested using the methods described in the next section. 

### 2.5. Harvest

Romaine lettuce heads were cut approximately three to four cm above the soil surface to reduce contamination from soil contact. In addition, lettuce head negative controls were randomly selected at the end of the lettuce field to minimize the chance of cross-contamination from inoculated animal fecal material. Harvested lettuce head samples were individually placed in 24 × 24 Bitran bags (Uline, Pleasant Prairie, WI, USA), transported to the laboratory on ice, and processed within 6 h. A change of gloves and disinfection of the harvesting knives with 70% alcohol between each head of lettuce was performed to reduce cross-contamination during harvesting and packaging. After lettuce samples were collected, the remnants of 23 fecal deposits in Experiment A were collected from the furrows and individually bagged into Whirl-Pak bags (Nasco, Ft. Atkinson, WI, USA) for bacterial enumeration. All negative control samples were collected and processed using the same procedures for the experimental samples.

### 2.6. Indicator E. coli Rif^r^ Most Probable Number (MPN) Determination (Experiment A) 

Using a modification of the procedures described in Atwill et al. [21] to quantify *E. coli* Rif^r^ per head of lettuce, 500 mL of PBS with 100 µg/mL of rifampicin were added directly into each bag and mixed vigorously with the lettuce head. The rationale for enumerating bacteria for the entire head of lettuce versus a subset of leaves was the observation in Atwill et al. [20] that bacterial contamination following foliar irrigation was highly heterogeneous across the head of lettuce, with the majority of bacteria concentrated on the lower outer and not inner leaves. Hence, sampling only a subset of leaves may not represent the actual cfu of bacteria per head of lettuce. For the high-concentration (HC) assay, 1 mL of the washate was transferred into the first three wells, then 10-fold serially diluted for four dilutions in a 12-well deep well reservoir containing TSB-Rif. In addition, a low concentration (LC) assay was used to broaden the range of detection limit. Briefly, washate was dispensed into Whirl-Pak bags containing irradiated TSB granules (Oxoid, Thermo Fisher, Waltham, MA, USA). This 4 × 3 MPN LC assay consisted of four different concentrations, in triplicate, of 100, 50, 10, and 5 mL aliquots containing a proportional concentration of TSB (30 g/1000 mL). TSB-Rif reservoirs and Whirl-Pak bags were incubated at 37 °C for 24 h with agitation of 100 rpm. The enriched washate was then channel streaked onto CHROMagar ECC (DGR international Inc. Mountainside, NJ, USA) supplemented with rifampicin (CHROMagar ECC-Rif). After incubation, blue colonies of indicator *E. coli* were identified as positive.

Each fecal deposit was collected from the furrows following irrigation, then added to 90 mL of PBS-rif and vigorously shaken to equally distribute the pathogen in each bag. One mL was pipetted into the first two wells and then 100-fold serially diluted for six dilutions in a 12-well deep well reservoir, incubated at 37 °C for 24 h with agitation, and streaked onto CHROMagar ECC-Rif plates. Additionally, 10 irrigation water samples collected directly from the sprinkler system were put in sterile plastic bottles to confirm the absence of *E. coli* resistant to rifampicin by adding 30 g of irradiated TSB granules (Oxoid, Thermo Fisher, Waltham, MA, USA) to one liter of water sample, and vigorously shaken to ensure an equal distribution within the samples. Water sprinkler samples were then incubated at 37 °C for 24 h and streaked onto CHROMagar ECC-Rif plates.

### 2.7. Indicator E. coli Rif^r^ Most Probable Number (MPN) Determination (Experiment B)

Using methods previously described by Chase et al. [28,34], a total of 500 mL of PBS with 100 µg/mL of rifampicin was added directly into each lettuce bag and mixed by vigorously shaking for 2–3 min. One mL of the washate was transferred to the first two wells of a a 12-well deep well reservoir containing 9 mL of TSB-Rif in each well. Five dilutions of 100-fold serial dilution were performed in the next ten wells of a 12-well deep well reservoir, containing 9.9 mL TSB-Rif. The reservoirs were shaken at 100 rpm and incubated overnight at 37 °C. The methods described by Chase et al. [28,34] were modified by streaking the enrichment from the reservoirs onto ChromAgar ECC-Rif to identify the presence of blue colonies, indicating the presence of *E. coli* Rif^r^. When the lettuce samples were negative for *E. coli* Rif^r^, an enrichment process was then carried out on the stored lettuce samples. Fifteen grams of irradiated TSB granules (Oxoid, Thermo Fisher, Waltham, MA, USA) were directly added to the stored samples and incubated at 37 °C overnight with agitation of 100 rpm. Ten µL of enrichment solution were streaked in ChromAgar ECC-Rif plates. After 24 h incubation, the amount of *E. coli* Rif^r^/lettuce head (MPN/ head) was recorded. 

### 2.8. Bacterial Concentrations and Confirmatory Test

Recovered bacterial concentrations were estimated using the MPN Calculator, Build 23 (http://member.ync.net/mcuriale/mpn/index.html) based on 95% confidence intervals. The combined detection limits for the LC and HC assays used in Experiment A ranged from 1.13 MPN/head to 3.51 × 10^7^ MPN/head of *E. coli* Rif^r^. The detection limits for Experiment B ranged from 341.6 MPN/head to 3.47 × 10^12^ MPN/head of *E. coli* Rif^r^. The raw MPN/mL or MPN/µL was extrapolated to a per head estimate by multiplying by the total washate volume, under the assumption that the washing techniques equally removed and homogenized the remaining *E. coli* Rif^r^ into the PBS-Rif washate. 

Twenty percent of the presumptive positive colonies were randomly selected from positive ChromAgar ECC plates. Isolates were stored at −80 °C and characterized using pulse field gel electrophoresis (PFGE) to ensure that recovered bacteria were the inoculated strain and not rifampicin-resistant background *E. coli* in the lettuce field. *Salmonella* Braenderup strain H9812 was used as a control according to the CDC PulseNet standard laboratory procedure [35]. After completion of PFGE, the gel was stained with ethidium bromide, the image captured by Gel/Chemi Doc (Bio-Rad, Hercules, CA, USA), and the genetic similarity determined between different strains using GelCompar II software (Applied Maths, Austin, TX, USA).

### 2.9. Statistical Analysis

For Experiment A, a negative binomial regression model was used to model the association between the concentration of *E. coli* Rif^r^ (C_t_) (MPN)/lettuce head) and various contributing factors such as the average distance from the fecal deposit to the four closest sprinklers (cm), distance from the lettuce to fecal deposit (cm), age of feces prior to irrigation (day), source of feces (chicken or rabbit), cumulative application of foliar irrigation (cm), wind aspect of lettuce (lettuce was upwind or downwind of fecal deposit), and mean wind speed (mph). An integer ID for each of the twenty-three clusters of eight adjacent heads of lettuce was used as a group or cluster variable in order to adjust *P*-values and CI’s for potentially correlated data within each lettuce cluster. Univariate models were first used to screen for potentially significant candidate variables for the multivariable regression model. A forward stepping algorithm was used to build the multivariable negative binomial regression model, with a *P*-value of ≤0.05 based on a likelihood ratio test for inclusion in the model. In order to improve interpretability, Figure 2A–C use log(C_t_/C_o_) as the y-axis variable, which is log of the ratio of the predicted value for C_t_ based on the regression model shown in Table 2 divided by C_o_, which is the mean MPN of *E. coli* Rif^r^ per 5 g fecal deposit for both chicken and rabbit feces (Table 1, 4.825 × 10^8^ MPN *E. coli* Rif^r^).

For Experiment B, a negative binomial regression model was also used to model the daily concentration of *E. coli* Rif^r^ (C_t_) following inoculation (C_t = 0_) of chicken and rabbit fecal slurries onto heads of Romaine lettuce for a 10-day trial, where t = daily collection event. The observed mean daily bacterial concentration per head of lettuce exhibited a non-monotonic relationship over time, which was modeled using a fully parameterized regression model that set the daily collection event as a categorical time variable. To account for heterogeneity in the initial bacterial concentrations of the different stock slurry solutions (chicken, rabbit), an offset variable for concentration at t = 0 was included in the negative binomial regression model. The dependent count variable was MPN/µL (i.e., the laboratory-observed or most native form), which was then used to extrapolate bacterial concentration to per head of lettuce by multiplying the model’s predicted outcomes by the total washate volume per head of lettuce. For statistical significance, two-sided hypothesis tests were used with a 95% confidence level.

## 3. Results

### 3.1. E. coli Rif^r^ Concentration on Romaine Lettuce after Irrigation (Experiment A)

As shown in Table 1, the initial concentration of *E. coli* Rif^r^ applied in 5 g of inoculated feces to each cluster of eight heads of lettuce ranged from 7.72 × 10^7^ to 2.00 × 10^8^ per g chicken feces and 1.48 × 10^7^ to 9.42 × 10^7^ per g rabbit feces. Depending on the number of days prior to irrigation and source of feces, the concentration of *E. coli* Rif^r^ in the fecal deposits following irrigation on the morning of day 5 varied greatly from the initial inoculum levels. Although some of this variability may be due to inherent differences in bacterial concentrations between successive aliquots of a bacterial stock solution, additional variability may have occurred due to multi-log bacterial growth or multi-log bacterial reduction relative to initial inoculum levels as evidenced by the large standard deviations in bacterial concentration per fecal deposit (Table 1). Due to this variability, there was no significant difference between the initial and final concentrations for chicken and rabbit feces (*P* > 0.05) despite the observation that the overall mean bacterial concentration post-irrigation had a higher value than before irrigation (Table 1). 

Seventy percent (126/180) of 180 mature heads of Romaine lettuce had detectable concentrations of indicator *E. coli* Rif^r^ after ~2 h of irrigation based on the high-concentration enumeration assay, with a mean concentration of 88,176 *E. coli* Rif^r^ per lettuce head (range was 178.3 to 549,475 MPN of *E. coli* Rif^r^). The remaining 54 negative samples out of the original 180 heads of lettuce were retested using a low concentration enumeration assay, with 88.9% (48/54) testing positive for indicator *E. coli* Rif^r^ and a mean concentration of 44.8 *E. coli* Rif^r^ per lettuce head (range was 1.10 to 197.60 MPN of *E. coli* Rif^r^). Combining the results from both HC and LC assays indicated that 96.7% (174/180) of lettuce heads had detectable concentrations of indicator *E. coli* Rif^r^, with a weighted mean (high and low concentration assays) of 64,933 *E. coli* Rif^r^ per positive lettuce head. None of the eight negative control heads of lettuce tested positive. Moreover, of the 35 positive samples (20% of all positives) that were used for DNA confirmation of the inoculum strain, all 35 isolates had an identical PFGE pattern to the inoculate strain. 

To evaluate the effectiveness of a 152.4 cm (5 ft) no-harvest buffer zone to reduce bacterial contamination of lettuce surrounding a fecal deposit, two additional lettuce samples were collected per cluster for a total of 47 heads of lettuce at an average distance of 136.09 ± 11.79 cm (4.47 ± 0.37 ft) from the fecal pat. Forty-four of these 47 heads of lettuce were within the 152.4 cm (5 ft) no-harvest zone (average distance 134.80 ± 8.62 cm (4.42 ± 0.29 ft); only three were just outside the 152.4 cm (5 ft) no-harvest zone (average distance 162.56 cm (5.33 ft)). Random sampling of heads of lettuce at the margins of this no-harvest zone did not allow perfect 152.4 cm (5 ft) distances between the 47 lettuce heads and the fecal deposits, which resulted in the observed variability in these distances. 

With respect to all 44 heads of lettuce within the 152.4 cm (5 ft) no-harvest zone, the median and mean (SD) bacterial concentration was 142.7 and 25,722 (± 88,281) *E. coli* Rif^r^ per lettuce head (range was 0 to 549,475 MPN *E. coli* Rif^r^). The large difference between the mean and the median is indicative of highly skewed data, with a few heads of lettuce containing very high bacterial levels. Five of these 44 (11%) lettuce heads were negative for *E. coli* Rif^r^. Among the 39 out of 44 (89%) lettuce heads testing positive, the median and mean (SD) concentration was 178.4 and 31,773 (± 94,694) *E. coli* Rif^r^ per lettuce head (range was 1.08 to 549,475 MPN *E. coli* Rif^r^). Regarding the three heads of lettuce just outside the 152.4 cm (5 ft) no-harvest zone, the median and mean (SD) concentration was 178.4 and 219.5 (± 205.3) *E. coli* Rif^r^ per lettuce head (range was 1.46 to 458 MPN *E. coli* Rif^r^), with all three heads of lettuce testing positive for *E. coli* Rif^r^.

### 3.2. Contributing Factors Associated and Not Associated with E. coli Rif^r^ Transfer from Feces to Lettuce (Experiment A)

Factors significantly associated with the concentration of *E. coli* Rif^r^/head of lettuce (log(MPN/lettuce head)), which is indicative of the transfer of bacteria from feces and/or furrow water down the gradient of the feces onto lettuce, include the mean distance from the four closest sprinklers to the fecal deposit (*P* = 0.002), distance from feces to each sampled head of lettuce (*P* = 0.037), cumulative amount of foliar irrigation applied to the block of lettuce (*P* = 0.037), and whether the head of lettuce was upwind or downwind of the fecal deposit (*P* = 0.043) (Table 2; Figure 2A–C). On the other hand, the age of the inoculated fecal material prior to irrigation and the source of feces (chicken or rabbit scat) were not significantly associated with transference of *E. coli* Rif^r^ from fecal deposits onto Romaine lettuce heads. Given the sample size of 180 heads of lettuce for this analysis, this model containing four factors is unlikely to be over fitted.

Based on the coefficients from the negative binomial regression model (see Table 2), each additional cm in the mean distance between the four closest sprinklers and the fecal deposit was associated with a ~1.4% increase (e^0^^.0137 × 1^ = 1.0138) in *E. coli* Rif^r^ concentration/head of lettuce. For each additional cm of distance between the fecal deposit and lettuce, the concentration of *E. coli* Rif^r^ decreased by ~1.1% (e^−^^0^^.0109 × 1^ = 0.989) or ~0.005-log decrease in *E. coli* Rif^r^ per additional cm of distance. Similarly, for each additional cm of applied irrigation water, the concentration of *E. coli* Rif^r^/lettuce head increased by 297% (e^1.0895 × 1^ = 2.973), or an approximate 0.47-log increase in bacterial deposition onto nearby heads of lettuce due to processes such as greater numbers of irrigation water droplets impacting the adjacent fecal deposit and/or impacting a larger surface area of ponded furrow water that then splashed onto the lettuce leaves. Finally, heads of lettuce located downwind of the fecal deposit had on average a 253% (e^0.9265^ = 2.526) higher concentration of *E. coli* Rif^r^ compared to lettuce located upwind, presumably due to wind enhancing the trajectory of fecal splash. Using this regression model to make tentative predictions, at the recommended 152.4 cm (5 ft) no-harvest buffer distance (152.40 cm (5.0 ft)) the predicted bacterial contamination per head of lettuce would be reduced by ~80% (e^−^^0^^.01089 × 152^^.4^ = 0.19) or 0.72-log reduction, compared to predicted bacterial levels on lettuce immediately adjacent to the point of fecal contamination in the furrow. This observation of a 0.72-log reduction provides support for the recommended food safety practice of a 152.4 cm (5 ft) no-harvest buffer when visible fecal deposits are observed in the furrow prior to harvest.

### 3.3. Environmental Inactivation of E. coli Rif^r^ in Chicken and Rabbit Fecal Slurries on Romaine Lettuce Surfaces (Experiment B)

Approximately 8.28 × 10^7^ and 1.07 × 10^8^ MPN *E. coli* Rif^r^ per lettuce head was applied on day = 0 (August 1) onto the adaxial surface of lettuce leaves using chicken and rabbit fecal slurry, respectively. During the 10-day trial, 100% of inoculated heads of lettuce had detectable levels of *E. coli* Rif^r^, with a range of 9.95 × 10^2^ to 3.47 × 10^12^ MPN per lettuce head with chicken feces (*n* = 96) and 1.12 × 10^2^ to 3.47 × 10^12^ MPN per lettuce head with rabbit feces (*n* = 95). Given that bacterial concentrations in excess of 4-log higher than the inoculum dose were observed on numerous heads of lettuce, these findings suggest that *E. coli* Rif^r^ was capable of replication on lettuce surfaces after the initial inoculation of ~10^8^ MPN/lettuce head. All *E. coli* isolates from a random subset (20%, *n* = 39) of positive heads of lettuce matched the PFGE pattern of the inoculum strain of *E. coli* Rif^r^. Forty-four negative controls and 10 samples of irrigation water all tested negative for the inoculum *E. coli* strain. 

Comparing the mean concentration of *E. coli* Rif^r^ per head of lettuce at day 0 compared to day 10 indicated that there was an overall multi-log reduction in bacterial levels during the 10-day trial, but substantial multi-log variation in bacterial concentration occurred within each day of the trial and there was significant departure from simple first-order inactivation processes for bacterial levels per lettuce head (Figure 3). For example, using the simple linear regression model shown in Figure 3 to estimate the overall 10-day mean inactivation rate, decimal reduction times (DRT) were 2.17 days (0.46 log reduction/day) for *E. coli* Rif^r^ in chicken fecal slurry on lettuce leaves and 2.50 days (0.40 log reduction/day) for *E. coli* Rif^r^ in rabbit fecal slurry on lettuce leaves under field conditions typical of the northwest section of the Salinas Valley, California (environmental data in the next section below). However, the use of negative binomial regression with sample date set as a categorical variable more accurately characterized these important departures from linear inactivation such as bacterial population growth up to several logs above inoculum levels immediately after inoculation and again after the mid-trial irrigation event on day 5, which would be impossible to characterize by simple linear regression without the use of third or fourth order polynomials (Figure 3).

### 3.4. Environmental Data (Experiment B)

Weather data were recorded by CIMIS from the North Salinas station (code 116). Based on readings from June, July, and August 2012, 24-h mean ambient temperatures were 12.92, 13.05, and 13.67 °C, with a relative humidity of 89.25, 87.80, and 89.72%, respectively. During the 10-day trial, there was no precipitation. Wind speed averaged over the month decreased from 8.23 in June, 7.27 in July, to 6.59 mph in August. The average soil temperature increased from 19.31 to 20.80 to 21.00 °C, respectively, during the three-month period.

## 4. Discussion

Following 2.5 h of foliar sprinkler irrigation for Experiment A, indicator *E. coli* Rif^r^ in animal fecal deposits in the furrow were transferred onto adjacent heads of lettuce, with 97% of sampled heads of lettuce having detectable levels of bacterial contamination. This high proportion of bacterial contamination was substantially larger than the 38% prevalence of *E. coli* O157:H7 on heads of Romaine lettuce observed in a similar trial using fecal deposits in the furrow and 2.5 h of foliar irrigation [20]. Although our current study used fecal material from laboratory rabbits and domestic chickens to simulate wild lagomorphs and avian species that forage in agricultural fields, these findings may have broader implications if the processes governing the transfer of fecal *E. coli* Rif^r^ onto the surface of Romaine lettuce and subsequent bacterial replication or inactivation rates are similar to what might be observed with *E. coli* in feces from other vertebrates such as feral pigs, raccoons, and deer. High inoculum levels were used in this field trial to allow for the quantification of bacterial transfer from fecal material onto adjacent lettuce heads and to also allow the estimation of environmental inactivation on the surface of lettuce leaves without excessive observations of zero counts that can downward bias both parameters (bacterial transfer, inactivation rate). The applicability of our results to situations in which *E. coli* concentrations or density of feces are less than those used in these experiments depends on whether transfer efficiency and inactivation rates are independent of initial concentration levels; we are unaware of published data relevant to these field studies that would suggest otherwise. Given this assumption of independence between transfer efficiency, inactivation rate, and bacterial concentration in feces, the observed DRT’s of 2.2 and 2.5 for *E. coli* in chicken and rabbit fecal slurries on lettuce leaves are supported by the findings of Moyne et al. [36], who found that warmer temperatures combined with high humidity enhanced survival (or reduced inactivation) of *E. coli* O157:H7 on lettuce leaves. However, inconsistent with Moyne et al. [36], bacterial populations in this study appear to have replicated to at least initial concentrations on day 0, especially following the mid-trial irrigation event. This observation of bacterial replication is consistent with findings observed by Chase et al. [34] where bacterial contamination, when delivered as a fecal slurry, provides a suitable environment for bacterial proliferation compared to contamination via irrigation water only (i.e., without fecal constituents). Environmental conditions during the 10-day trial in Salinas Valley, CA, USA were moderately warm (mid-13 °C for 24-h average ambient temperature), with average relative humidity of almost 90%. The persistence of *E. coli* Rif^r^ might have been greater given the observation that *E. coli* survival can be enhanced on injured iceberg lettuce compared to uninjured lettuce under overhead sprinkler irrigation with contaminated water [37].

An important objective of Experiment A was to evaluate the no-harvest buffer distance of 152.40 cm (5.0 ft) from the point of fecal deposition as a food safety policy that can reduce the likelihood of produce contamination due to animal defecation in agricultural fields [33]. Under the conditions of this field experiment, the longest documented distance of *E. coli* Rif^r^ transfer from the point of fecal contamination to adjacent heads of lettuce was 162.56 cm (5.33 ft), with a range of 109.22 to 162.56 cm (3.38 to 5.33 ft). We cannot rule out that bacterial transfer over these longer distances was due solely to fecal splash, but instead, the result of secondary splash from droplets of irrigation water impacting the surface of ponded water in the furrow down slope of the fecal deposit. Regardless of the mechanism of contamination, 97% of the heads of lettuce had *E. coli* Rif^r^ contamination when the original source of bacteria was the fecal deposit. The estimated ~0.005-log (1.1%) reduction in *E. coli* Rif^r^ concentration per head of lettuce for each additional cm of distance between the fecal deposits and lettuce suggests that implementing the food safety recommendation of the no-harvest buffer distance of 152.40 cm (5.0 ft) could substantially reduce the level of *E. coli* contamination on produce from overhead sprinkler irrigation given that in-field fecal deposits are present.

Regarding factors influencing the transfer of bacteria onto heads of lettuce for Experiment A, the cumulative application of foliar irrigation, average distance from feces to the four closet sprinklers, distance between fecal material and lettuce, and whether the head of lettuce was upwind or downwind of fecal deposit were significantly associated with the concentrations of *E. coli* Rif^r^/lettuce head, whereas the age of the inoculated fecal material and source of feces were not significant variables on the transference of *E. coli* Rif^r^ from fecal materials to Romaine lettuce. An experimental study by Atwill et al. at the same geographical location found similar factors associated with bacterial transference onto heads of lettuce: (e.g., average distance from feces to the four closet sprinklers, distance between fecal material and lettuce, aspect of wind relative to beds of lettuce), but in contrast to the present study, it also found that the age of fecal material before irrigation was significantly associated with concentrations of *E. coli* O157:H7 per head of Romaine lettuce [20]. Even though the source of feces was not associated with the concentration of *E. coli* Rif^r^ per head of lettuce in our study, a study by Mishra et al. indicated that rainfall, soil, irrigation, and source of feces (cattle and wild feral pigs) were associated with the survival of *E. coli* O157:H7 related to lefty green contamination based on a dynamic system model [38]. 

The cumulative amount of irrigation water applied during the 2.5 h was positively associated with bacterial concentration on heads of lettuce. Natural variation in the application of irrigation water across the study plots, which ranged from 0.2 cm to 2.4 cm/cluster of evaluated lettuce, allowed this effect on bacterial transfer to be modeled (e.g., each additional cm of overhead irrigation water increased the concentration of *E. coli* Rif^r^ per lettuce head by 297% or 0.47-log). This positive association suggests that the process(es) of bacterial deposition onto the surface of lettuce during foliar irrigation overwhelm the competing process(es) of removal (e.g., shearing, detachment), resulting in a relative increase in bacterial concentrations as a function of foliar irrigation (within the constraints of our study design). One can speculate that at some point, microbial transfer becomes bacterial source-limited following excessive amounts of irrigation water [39], or that a different distribution of droplet size will result in a different association between amount of foliar irrigation and bacteria transfer. This finding of a positive association between bacterial transfer and amount of foliar irrigation is supported by a study in the Netherlands where the transfer of *E. coli* O157:H7 from manure-amended soil onto lettuce occurred by rainfall droplets causing soil-manure splash, with the magnitude of transfer related to droplet size and rainfall intensity [40]. Pielaat and Van den Bosch [41] developed a stochastic model of pathogen transference that predicted the spatial spread of plant pathogen spores was driven by dispersal from rainfall, whereby higher intensity rainfall resulted in more splash, which in turn led to greater pathogen dispersal. Furthermore, an experimental field trial from Chase et al. indicated multi-log regrowth of *E. coli* O157:H7 on Romaine lettuce was observed after foliar irrigation [34]. Finally, in an experimental study using irrigation water containing 10^7^ CFU *E. coli* O157/mL, 90% (29/32) of lettuce plants were contaminated with *E. coli* O157:H7 after being spray irrigated, while only 19% (6/32) of plants were contaminated after using surface irrigation [25]. These studies, in addition to Atwill et al. and Weller et al. [20,30], indicate that overhead sprinkler or foliar irrigation, similar to natural precipitation, can transfer bacteria either directly from the fecal deposit onto the heads of lettuce, or indirectly by contaminated furrow water splashing onto the lettuce, with possibly droplet size and amount of foliar irrigation (present study) positively correlated to microbial transfer.

For Experiment A, each additional cm in the mean distance between the four sprinklers and the fecal deposit was associated with a 1.4% increase (or 0.006-log) in *E. coli* concentration per head of lettuce. Although somewhat counter intuitive, given that the four impact sprinklers were set at the corners of a square with 9.12 m edges, the mean distance was lowest in the middle of the square and increased as one approached any of the sprinkler risers (6.44 m in the middle, 7.6 m adjacent to any riser). Therefore, transfer of *E. coli* Rif^r^ onto the heads of lettuce increased as one moves from the center of the square plot outward to the edges. During the irrigation trial, we informally observed that irrigation water with the largest droplet size and presumably highest kinetic energy impacted several feet inside the edge of each plot from the opposing risers, which would maximize the erosive force of irrigation water on the fecal deposit and ponded furrow water and thereby facilitate bacterial dispersal onto nearby heads of lettuce. In other words, it may be the impact from irrigation water from one or more of the furthest three risers that cause a significant proportion of bacterial transfer. Kincaid et al. [42] observed that smaller droplets of water tend to fall closer to the sprinkler head that they were emitted from, while larger droplets fell further away from the sprinkler head and caused the wetted outer edge of the irrigation circle; hence, lettuce clusters with a larger mean distance from all four sprinklers were in fact close to one of these sprinklers and being impacted by one or more of the other three sprinklers leading to higher bacterial transfer. In addition, we observed that risers at the edge of field next to the access road had irrigation guards installed to prevent saturation of the road. For each riser rotation, these guards redirected the entire flow of water as a short arch of dense water, likely causing additional bacterial transfer from fecal deposits onto adjacent heads of lettuce. Fonseca et al. [43] found that under different irrigation systems, a greater probability of *E. coli* contamination was observed from overhead sprinkler irrigation compared to subsurface drip and surface furrow systems, owing to the direct contact of contaminated water droplets with edible parts of plant tissue.

The motivation for this study was to better understand how wildlife defecation in fields of produce can result in pathogen contamination of produce grown in beds. The reported prevalence of *E. coli* O157:H7 contamination in avian species (e.g., duck, goose, gull, pigeon, swan, turkey, and deer) is usually lower than 10% in the U.S. [16,44,45]. Although high concentrations of *E. coli* O157:H7 contamination have been reported in vertebrate sources such as cattle [46,47], two epidemiological surveys done in California on predominately rangeland cow-calf beef operations found that the overall prevalence was less than 5%, with many herds persistently testing negative over a 2- to 3-year time frame [48,49]. This low prevalence suggests that range beef cattle in the proximity of produce fields, when excluded from produce fields by fencing, are not likely the primary reservoir for *E. coli* O157:H7 for all agricultural or rural systems in the U.S. Instead, and especially in regions where the prevalence of *E. coli* O157:H7 is consistently low in local cattle herds, it may be that infected wildlife that defecate directly into produce fields function as a more efficient source of produce contamination, particularly for mature crops utilizing foliar irrigation near harvest.

Wild birds, deer, and feral swine have been suggested as potential wildlife sources of *E. coli* O157:H7 in the pre-harvest production environment due to their habitat and proximity to agricultural fields and livestock production systems [16,50,51]. After infected wildlife defecate into produce fields, various irrigation methods can mobilize and disperse the microbes in fecal material and cause cross-contamination in agricultural products as shown in the current Experiment A and related studies [20,28,30,31]. Once on the lettuce leaf, the mean DRTs for *E. coli* Rif^r^ were 2.2 to 2.5 days when the bacterial inoculum was delivered as a fecal slurry under these agricultural and environmental conditions. Interestingly, the chicken slurry DRT in Experiment B of this field study was similar to a DRT value of 1.9 days at 37 °C observed for *E. coli* O157:H7 in chicken feces [52]. 

In Experiment A of this study, we used 5 g of chicken and rabbit fecal slurry containing 9.65 × 10^7^
*E. coli* Rif^r^ CFU/g feces in an effort to simulate *E. coli* O157:H7 contamination from a super-shedder animal. Super-shedder animals such as feral pigs and cattle by definition have high concentrations of the target bacteria in their feces. For example, cattle have been shown to shed up to 8.4 log of *E. coli* O157:H7 per g feces [53,54], and concentrations of 7 log CFU per g feces of *E. coli* O157:H7 have been observed in feral pigs [55]. Relative to the 4.825 × 10^8^
*E. coli* Rif^r^ CFU inoculum dosage per fecal deposit, we observed an average of 6.49 × 10^4^
*E. coli* Rif^r^ CFU per positive head of lettuce. Calculating the ratio of *E. coli* Rif^r^ per head of lettuce to the bacterial load in the adjacent fecal deposit gives an indication of the proportion of fecal bacterial load that transferred onto nearby heads of lettuce; for this study we observed an average of 0.0135% (6.49 × 10^4^/4.825 × 10^8^) of the total bacterial load was transferred onto these positive heads of lettuce relative to the original inoculum dosage, but with considerable variability around the mean. Presumably, if the *E. coli* concentration in wildlife feces was substantially less than this super-shedder concentration used in our experiment, one would observe proportionally less *E. coli* on each head of lettuce. In a previous study at this location, Atwill et al. [20] found a similar value of 0.00573% of the original bacterial load of 1.29 × 10^8^ CFU per 5 g fecal deposit transferred to nearby heads of lettuce. This suggests that if we had spiked our fecal deposits with, for example, only 10,000 *E. coli* CFU/g feces, on average, only ~7 CFU would be transferred to nearby heads of lettuce and even less for heads at the no-harvest 152.40 cm (5.0 ft) buffer distance. This line of reasoning assumes that the amount of bacterial transference is proportional to bacterial concentration in the fecal deposit and factors such as density of wildlife feces on the soil surface.

Studies on the persistence of *E. coli* O157:H7 on produce are limited, with even fewer studies focusing on this bacterium in a fecal slurry located on lettuce leaves [28,31,34,56,57,58]. A recent study conducted in the Salinas Valley examined the inactivation of an attenuated *E. coli* O157:H7 strain on Romaine lettuce inoculated in a rabbit fecal slurry and found a mean 2.3 log reduction at 92 h (3.8 days), which equates to 0.6 log MPN/day [28]. In comparison, we observed, during Experiment B, a slightly longer survival time in the present study using *E. coli* Rif^r^ on Romaine lettuce, with rates of inactivation ranging from 0.40 to 0.46 log MPN/day. Weller et al. [31] conducted a similar field trial in New York using indictor *E. coli* Rif^r^ and reported a 0.52 log MPN/day inactivation rate, which indicated that the strain behaved similarly following inoculation in a fecal slurry onto lettuce leaves in two different regions, central coastal California and upstate New York. An earlier study found that *E. coli* O157:H7 concentrations in cattle slurry rapidly decreased and reached undetectable levels within 10 days post-inoculation, with an initial concentration of 10^6^ CFU/mL, consistent with a 0.6 log reduction/day [56]. Another study reported that the persistence of *E. coli* O157:H7 in feces varied depending on temperature and possibly ammonium levels in feces, where the survival range was from two days to five weeks; the inactivation rate was highest at 37 °C, followed by 20 °C and 4 °C [57]. Potentially influencing our calculations of DRT across the 10-day duration was the proportion of heads of lettuce exhibiting bacterial regrowth on day 6 following irrigation on day 5 (Figure 3), likely extending the persistence of *E. coli* on these heads of lettuce and enhancing the food safety risk. 

Given the large amount of intra-day variation in the bacterial concentration per head of lettuce during Experiment B (i.e., in excess of 5-log in some cases), combined with the observation of substantial bacterial regrowth following irrigation, we are concerned that over reliance on multi-day linear trends like the SLR models shown in Figure 3 for log-normal data and the common use of daily arithmetic means (or even worse, geometric means) to represent food safety risk from these commodities can dramatically over-smooth and underestimate the actual risks experienced when consuming these commodities, especially if someone is unfortunate enough to consume lettuce from a head exhibiting multi-log bacterial replication in the field, as can be readily seen in Figure 3. Unfortunately, at this time, no highly accurate predictive models exist to identify a priori which heads of lettuce will experience multi-log replication versus multi-log reduction despite the contributing factors shown in Table 2. Additional mechanisms of bacterial persistence remain to be discovered and put to use for improving food safety from these produce commodities. 

## 5. Conclusions

This study characterized the transfer of *E. coli* from animal feces and/or furrow water onto adjacent heads of lettuce during foliar irrigation, and the subsequent survival of bacteria on the adaxial surface of lettuce leaves. Analysis of these data identifying factors influencing the amount of indicator *E. coli* Rif^r^ transferred to lettuce from the point of fecal contamination under sprinkler irrigation. Inferences from these results suggest that when contaminated fecal deposits are present, extending the time period between foliar irrigation and harvest, along with a 152.4 cm (5 ft) no-harvest buffer zone may substantially reduce the level of bacterial contamination on harvested lettuce.

## Figures and Tables

**Figure 1 microorganisms-07-00408-f001:**
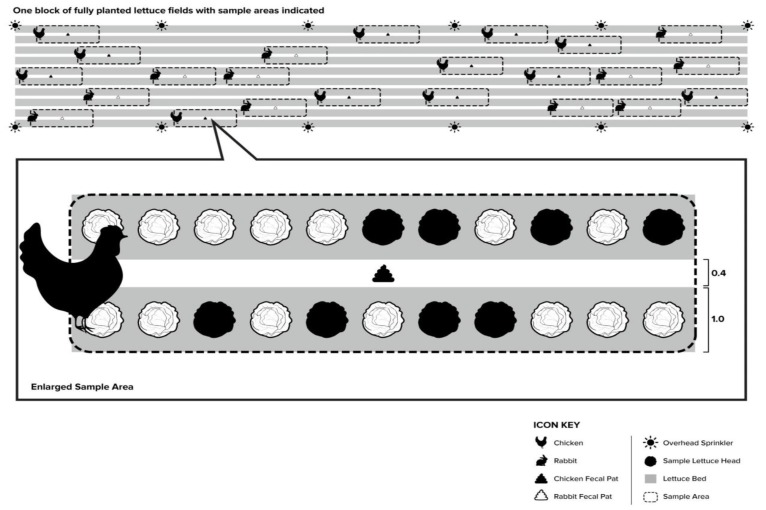
Field layout for one cluster of mature Romaine heads used in the *E. coli* transfer experiment. Each cluster was composed of eight heads of mature Romaine lettuce equally divided between two adjacent beds on each side of the furrow and a fecal deposit placed in the middle of the furrow.

**Figure 2 microorganisms-07-00408-f002:**
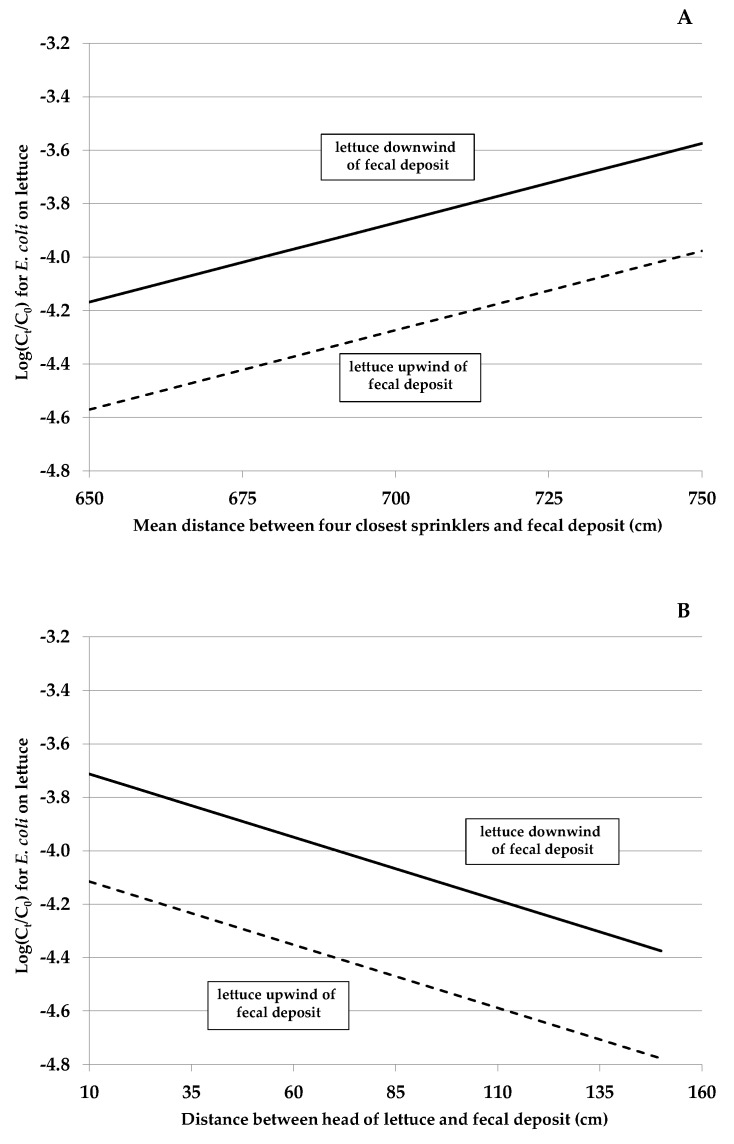
The predicted effect of (**A**) mean distance from the four closest sprinklers to the fecal deposit, (**B**) distance between head of lettuce and the fecal deposit, and (**C**) cumulative application of foliar irrigation on log(C_t_/C_o_). C_t_ represents the predicted bacterial concentration per head of lettuce under the stated conditions based on the negative binomial regression model shown in Table 2, and C_o_ is the mean number of inoculated *E. coli* Rif^r^ per 5 g of chicken or rabbit feces (4.825 × 10^8^ MPN).

**Figure 3 microorganisms-07-00408-f003:**
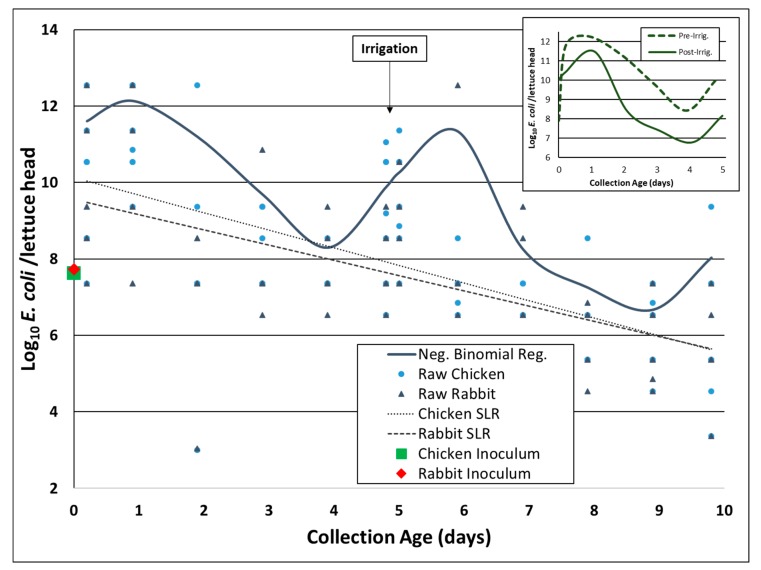
Experiment B. Daily concentration of indicator *E. coli* Rif^r^ on Romaine heads of lettuce (logMPN/head of lettuce) modeled using simple linear regression (SLR) and negative binomial regression: Chicken (*n* = 96), and Rabbit (*n* = 95). Bacteria inoculated as a fecal slurry onto lettuce heads on day 0, with foliar irrigation occurring on day 5. Heads of lettuce were harvested every day, and also immediately before and after irrigation on day 5. The subplot compares the difference in the bacterial inactivation rate by setting pre- and post-irrigation samples on the same timeline.

**Table 1 microorganisms-07-00408-t001:** Experiment A. Mean concentration of *E. coli* Rif^r^ in 5 g of chicken and rabbit feces placed in the furrow up to four days prior to irrigation day (day 0) compared to *E. coli* Rif^r^ concentrations measured immediately after irrigation.

Days Prior to Irrigation	The Mean Bacterial Inoculum as MPN/g, (Standard Deviation)			
	Chicken Feces (*n* = 3)		Rabbit Feces (*n* = 3)	
	Initial Concentration *	After Irrigation	Initial Concentration *	After Irrigation
Day −4	1.06 × 10^8^	2.01 × 10^8^ (1.92 × 10^8^)	1.48 × 10^7^	2.58 × 10^9^ (3.00 × 10^9^)
Day −2	7.72 × 10^7^	2.33 × 10^11^ (3.25 × 10^11^)	9.42 × 10^7^	2.49 × 10^11^ (3.15 × 10^11^)
Day −1	2.00 × 10^8^	2.33 × 10^11^ (3.25 × 10^11^)	5.44 × 10^7^	4.17 × 10^9^ (3.00 × 10^9^)
Day 0	1.69 × 10^8^	2.58 × 10^9^ (3.00 × 10^9^)	5.65 × 10^7^	4.64 × 10^11^ (3.24 × 10^11^)
Overall mean	1.38 × 10^8^	1.17 × 10^11^ (2.79 × 10^11^)	5.50 × 10^7^	1.79 × 10^11^ (2.65 × 10^11^)

* Initial concentration was determined from the concentration in the fecal slurry stock solution.

**Table 2 microorganisms-07-00408-t002:** Experiment A. Negative binomial regression model for in-field contributing factors associated with the transfer of indicator *E. coli* Rif^r^ onto heads of mature Romaine lettuce (MPN/lettuce head) from animal feces placed in the furrow adjacent to beds of lettuce.

In-Field Factor	Coefficient	95% C.I. *	*P*-Value *
Mean distance from feces to four sprinklers (cm)	0.0137	0.0048 to 0.022	0.002
Distance from feces to head of lettuce (cm)	−0.0109	−0.021 to −0.0007	0.037
Cumulative application of foliar irrigation (cm)	1.0895	0.068 to 2.11	0.037
Head of lettuce downwind of the fecal deposit ^⊥^	0.9265	0.029 to 1.82	0.043
Intercept	−0.0328	−7.35 to 6.69	0.927

* 95% C.I. and *P*-values were adjusted for intra-group correlation; group ID set as twenty-three clusters of eight adjacent heads of lettuce, with 5 g of chicken or rabbit feces containing an average of 4.825 × 10^8^
*E. coli* Rif^r^ MPN placed in the center of each cluster (see Figure 1). ^⊥^ Referent category was lettuce upwind of the fecal deposit.

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
