# Peer review of "Experimental In-Field Transfer and Survival of Escherichia coli from Animal Feces to Romaine Lettuce in Salinas Valley, California"

_microorganisms, 2019, doi:10.3390/microorganisms7100408_

Round 1

Reviewer 1 Report

This manuscript describes a set of experiments to address the proposed buffer zone in growing leafy greens to reduce the risk of contamination associated with wildlife. While many assumptions have been made, specific data is lacking. The studies were performed with an important indicator strain used by many studies. Overall the paper is well written and important to the field. 

In abstract, line 18, the authors should move “to quantify” into the points 1 and 2 for clarification.

In abstract line 27, “ponded furrow water”? Is this pooled water?

In section 2.1, were the irrigation practices ones normally done for lettuce cultivation? This should be stated if so.

Would sampling lettuce leaves be more sensitive than sampling lettuce heads? Perhaps the authors could add a sentence on why lettuce heads were sampled and the ease of which bacteria could be recovered from heads. It seems that bacteria would be more easily unrecovered and attach to the lettuce leaves.

Experiments A and B should continue to be described as such in the figures and discussion.

Was there regrowth observed that could have been due to or associated with environmental conditions?

Author Response

Thank you for your constructive review.

In abstract, line 18, the authors should move “to quantify” into the points 1 and 2 for clarification.

We made the revision as requested.

In abstract line 27, “ponded furrow water”? Is this pooled water?

We deleted the term “ponded” so simply interpretation of the sentence.

In section 2.1, were the irrigation practices ones normally done for lettuce cultivation? This should be stated if so.

Yes, we utilized standard irrigation practices and standard commercial irrigation equipment. We added terminology to clarify this point, “These growing and irrigation practices along with the field design are typical of commercial operations in the region.”

Would sampling lettuce leaves be more sensitive than sampling lettuce heads? Perhaps the authors could add a sentence on why lettuce heads were sampled and the ease of which bacteria could be recovered from heads. It seems that bacteria would be more easily unrecovered and attach to the lettuce leaves.

Our concern with sampling a set of leaves rather than the entire head was our concern that the spatial pattern of bacterial contamination was not uniform across the head, but rather highly heterogeneous. Therefore, we would generate highly variable data from sampling leaves per head rather than the entire head. This concern was based on a related study conducted in 2011 where we observed bacterial contamination from foliar irrigation was primarily limited to outer lower leaves versus the interior section of the lettuce head (Atwill et al., 2015 doi:10.4315/0362-028X.JFP-14-277). Hence, in order to have a more valid count of bacterial contamination we tested the entire head of lettuce and report our results as cfu per lettuce head.

The following two sentences were added to section 2.6: “The rationale for enumerating bacteria for the entire head of lettuce versus a subset of leaves was the observation in Atwill et al. [20] that bacterial contamination following foliar irrigation was highly heterogeneous across the head of lettuce, with the majority of bacteria concentrated on the lower outer and not inner leaves. Hence, sampling only a subset of leaves may not represent the actual cfu of bacteria per head of lettuce.”

Experiments A and B should continue to be described as such in the figures and discussion.

We added references throughout the discussion to experiment A or B as needed, in addition we labeled the figure and table legends as experiment A or B.

Was there regrowth observed that could have been due to or associated with environmental conditions?

We speculate in the results (lines 295 to 297 in original manuscript) and discussion (lines 440 to 443 in original manuscript) that bacterial regrowth likely occurred in some of the samples which may explain some of the larger standard deviation values in Table 1 and the high levels of variability observed in Figure 3.

Reviewer 2 Report

Note that when indicating page numbers, I do not use the “# of 20” page indicators shown at the top, since these start on page 6. 

The abstract can benefit from the following:

A declaration that a randomized controlled trial was used. A clearer, simpler statement of the objective of the study, such as the first sentence under Conclusions, or the “motivation” statement on page 15, line 523.

Communication of quantitative aspects can benefit from the following: 

You describe a cumulative volume of irrigation measured via rain gauges. For many scientists, the measurement of volume is typically in a cubic length rather than in merely length.  For your first mention of the gauge and metric, briefly describe how a unit of length (cm) equates to a volume.  If you decide to modify your metric or units, follow through in the rest of the paper. Page 8, Line 270: Aspects of the paragraph seem to describe results from a regression.  Consider reserving them for the Results portion of the paper. Table 1: The negative signs indicating days before irrigation may be confusing.  Consider modifying the column heading to Days Prior to Irrigation and using positive numbers (i.e., 4, 2, 1, 0).  Use the term “mean” rather than “average”.  When describing percentages and indicating samples out of total samples (e.g., 44/47), consider whether some provide repetitive information and eliminate, if needed. For example on page 9, Line 315, the text describes “Fourty-four (44/47) of these 47 heads of lettuce”.  For much of the manuscript, the mean is used as a measure of central tendency. On page 9, line 321 there is the inclusion of the mean and media.  Do median values add anything here?  Are they used in a nonparametric test or within a table?  I would suggest being consistent – either using median and mean throughout or choosing one or the other.  Verify your conversions from regression coefficients to percent change on pages 11 and the top of page 12. The mathematics applied does not seem consistent.  If values change, then follow through with these changes in the rest of the manuscript.  Plots in Figure 2 may be unnecessary and misleading. Typically if points and regression lines are presented, readers likely assume that points are collected data and the lines are an attempt at fitting trends.  I feel rather than using plots, your results can be more easily communicated with the typical language used for regression coefficients (e.g., with a 1 unit change in beta we would expect an x unit change in the outcome).  There should likely be some mention of the limitation of potentially ‘overfitting’ data. If quantitative values are repeated, try to be consistent. For example on page 16, line 542, the DRT is communicated as 2.2. 

Language can be clarified by addressing the following: 

The term “transfer coefficient” is only used twice in the document. This term has a specific definition in exposure science (i.e., specifically for exposure modeling to contaminants including microbes) and may confuse the reader.  If the goal is indeed to compute transfer coefficients, it may be useful to use this term more often in the paper, highlight the quantity of these coefficients in tables, and reinforce the determined values in the conclusions.  If not then use other terms such as “cross-contamination”.  Review the manuscript for relatively large spaces between words (e.g., page 3, line 137). Some of this may be due to the justification.  Review the manuscript for consistent justification. For example it appears that text may be centered on the bottom of page 12 and top of page 13.  Generally review the manuscript for spelling errors. For example: page 6, line 172 page 14, line 436 page 14, line 470 Check grammar, word choice, and complex sentences: Page 9, line 301 Page 9, line 306 Page 9, lines 317-320 Page 12, lines 385-389 Page 15, lines 517-519 Page 15, lines 525-526 (examples of wild birds) Page 16, lines 552-557 Page 16, lines 561-565 Page 16, lines 567-570

General scientific question:  Was velocity of water streams measured or could it be measured?  Do you think this could be a useful explanatory variable?  

Author Response

Thank you for your constructive review.

The abstract can benefit from the following: A declaration that a randomized controlled trial was used. A clearer, simpler statement of the objective of the study, such as the first sentence under Conclusions, or the “motivation” statement on page 15, line 523.

As requested we edited the first sentence of the abstract to closely follow the first sentence of the conclusion, “These randomized controlled trials characterized transfer of E. coli from animal feces and/or furrow water onto adjacent heads of lettuce during foliar irrigation, and the subsequent survival of bacteria on the adaxial surface of lettuce leaves.”

 Communication of quantitative aspects can benefit from the following:  You describe a cumulative volume of irrigation measured via rain gauges. For many scientists, the measurement of volume is typically in a cubic length rather than in merely length.  For your first mention of the gauge and metric, briefly describe how a unit of length (cm) equates to a volume.  If you decide to modify your metric or units, follow through in the rest of the paper.

As requested, we edited lines 119 to 121 to be, “After harvesting all lettuce for E. coli determination, the cumulative volume of foliar irrigation (cm3) was measured using two rain gauges placed within each cluster of eight lettuce heads, which was simplified for the remainder of this paper as cm.”

Page 8, Line 270: Aspects of the paragraph seem to describe results from a regression.  Consider reserving them for the Results portion of the paper.

We kept the text as originally written given that the entire paragraph is a description of the statistical methodology we used for regression model fitting and reported the results of the model.

Table 1: The negative signs indicating days before irrigation may be confusing.  Consider modifying the column heading to Days Prior to Irrigation and using positive numbers (i.e., 4, 2, 1, 0).  Use the term “mean” rather than “average”. 

We edited Table 1 as requested.

When describing percentages and indicating samples out of total samples (e.g., 44/47), consider whether some provide repetitive information and eliminate, if needed. For example on page 9, Line 315, the text describes “Fourty-four (44/47) of these 47 heads of lettuce”. 

As requested, we deleted two examples of parentheses containing the ratio.

For much of the manuscript, the mean is used as a measure of central tendency. On page 9, line 321 there is the inclusion of the mean and median.  Do median values add anything here?  Are they used in a nonparametric test or within a table?  I would suggest being consistent – either using median and mean throughout or choosing one or the other. 

In this section of the manuscript we reported both the mean and median due to the dramatic difference between the two values since there were a small number of observations with very high values, which then skews the mean. We added a sentence (lines 330-331 in revised manuscript) to this section of the results to clarify that we reported both mean and median due to this skewness. “The large difference between the mean and the median is indicative of highly skewed data, with a few heads of lettuce containing very high bacterial levels.”

Verify your conversions from regression coefficients to percent change on pages 11 and the top of page 12. The mathematics applied does not seem consistent.  If values change, then follow through with these changes in the rest of the manuscript. 

I double checked the conversion from regression coefficient to percent change and did not find any errors. The use of exponential regression models like negative binomial or Poisson are by definition multiplicative models when showing results not in log scale but in the native scale, hence, interpretation of the regression coefficients is multiplicative.

Plots in Figure 2 may be unnecessary and misleading. Typically if points and regression lines are presented, readers likely assume that points are collected data and the lines are an attempt at fitting trends.  I feel rather than using plots, your results can be more easily communicated with the typical language used for regression coefficients (e.g., with a 1 unit change in beta we would expect an x unit change in the outcome).

We included the markers only for visual clarity. As requested, we deleted the markers and only show the regression line. Because these models are exponential models and interpreting coefficients using multiplicative language can be confusing (as explained in preceding point), it can be hard for some readers to interpret the model’s coefficients from a table alone. Therefore, we would prefer to include interpretative figures for ease of interpretation.

There should likely be some mention of the limitation of potentially ‘overfitting’ data.

We added a sentence to section 3.2 of the model results indicating that a sample size of 180 for a model with just four factors is unlikely to be overfitted.

If quantitative values are repeated, try to be consistent. For example on page 16, line 542, the DRT is communicated as 2.2.

As requested, we edited the earlier DRT value of 2.17 to be 2.2 in the line you are referencing.

Language can be clarified by addressing the following:

The term “transfer coefficient” is only used twice in the document. This term has a specific definition in exposure science (i.e., specifically for exposure modeling to contaminants including microbes) and may confuse the reader.  If the goal is indeed to compute transfer coefficients, it may be useful to use this term more often in the paper, highlight the quantity of these coefficients in tables, and reinforce the determined values in the conclusions.  If not then use other terms such as “cross-contamination”.

As requested we revised the two occurrences of “transfer coefficient” (last paragraph of the introduction, first paragraph of discussion) to be “transfer of E. coli” and “bacterial transfer” respectively.

Review the manuscript for relatively large spaces between words (e.g., page 3, line 137). Some of this may be due to the justification.  Review the manuscript for consistent justification. For example it appears that text may be centered on the bottom of page 12 and top of page 13.

These errors were fixed.

Generally review the manuscript for spelling errors. For example: page 6, line 172 page 14, line 436 page 14, line 470 Check grammar, word choice, and complex sentences: Page 9, line 301 Page 9, line 306 Page 9, lines 317-320 Page 12, lines 385-389 Page 15, lines 517-519 Page 15, lines 525-526 (examples of wild birds) Page 16, lines 552-557 Page 16, lines 561-565 Page 16, lines 567-570

We have revised each of the referenced sentences for clarity and syntax.

General scientific question:  Was velocity of water streams measured or could it be measured?  Do you think this could be a useful explanatory variable?

We did not measure the velocity of irrigation water being emitted from the sprinkler, but one would assume that there is a wide variety of droplet velocities given the range of impact in the studied lettuce plots.